# Identification of an Optimal TLR8 Ligand by Alternating the Position of 2′-O-Ribose Methylation

**DOI:** 10.3390/ijms231911139

**Published:** 2022-09-22

**Authors:** Marina Nicolai, Julia Steinberg, Hannah-Lena Obermann, Francisco Venegas Solis, Eva Bartok, Stefan Bauer, Stephanie Jung

**Affiliations:** 1Institute for Immunology, Philipps-University Marburg, 35043 Marburg, Germany; 2Institute of Cardiovascular Immunology, University Hospital Bonn, University of Bonn, 53127 Bonn, Germany; 3Institute of Experimental Haematology and Transfusion Medicine, University Hospital Bonn, University of Bonn, 53127 Bonn, Germany

**Keywords:** TLR7, TLR8, 2′-O-ribose-methylation, RNase T2, immune activation

## Abstract

Recognition of RNA by receptors of the innate immune system is regulated by various posttranslational modifications. Different single 2′-O-ribose (2′-O-) methylations have been shown to convert TLR7/TLR8 ligands into specific TLR8 ligands, so we investigated whether the position of 2′-O-methylation is crucial for its function. To this end, we designed different 2′-O-methylated RNA oligoribonucleotides (ORN), investigating their immune activity in various cell systems and analyzing degradation under RNase T2 treatment. We found that the 18S rRNA-derived TLR7/8 ligand, RNA63, was differentially digested as a result of 2′-O-methylation, leading to variations in TLR8 and TLR7 inhibition. The suitability of certain 2′-O-methylated RNA63 derivatives as TLR8 agonists was further demonstrated by the fact that other RNA sequences were only weak TLR8 agonists. We were thus able to identify specific 2′-O-methylated RNA derivatives as optimal TLR8 ligands.

## 1. Introduction

As the first line of host defense, the innate immune system recognizes molecular danger signals and subsequently triggers signal transduction and secretion of interferons (IFN) and proinflammatory cytokines [1,2]. This function is enabled by pattern-recognition receptors (PRRs), which recognize pathogen-associated molecular patterns (PAMPs) characteristic of a particular group of pathogens. Among these PAMPs are bacterial and viral nucleic acids, which are recognized not only by their structure and composition, but also by their intracellular localization [3]. For example, single-stranded RNA (ssRNA) in human endosomes is recognized by the two related Toll-like receptors (TLRs) 7 and 8, inducing diverse cytokine patterns in different cell types and species. In humans, TLR7 is strongly expressed in plasmacytoid dendritic cells (pDCs) and B cells, where its activation leads to B cell activation and IFN-α release from pDCs [3,4,5,6,7]. In contrast, TLR8 is primarily expressed in the myeloid compartment and its activation induces proinflammatory cytokine and IFN-β release [3,4,5,8]. In mice, TLR7 also functions similarly in pDCs and B cells [9,10,11]. However, it is further expressed in the myeloid compartment, where it induces a similar cytokine profile to human TLR8 [12,13]. In contrast, murine TLR8 does not induce cytokine release, and its function remains largely unknown [9,10].

Early reports stating that TLR7 and TLR8 preferentially recognize uridine-rich sequences were followed by structural analyses showing that both receptors bind single nucleosides and short ssRNA sequences at two distinct binding sites [14,15,16,17,18,19]. Recent studies have shown that the activities of endosomal ribonuclease (RNase) T2 and RNase 2 are essential for TLR8 activation; RNase activity generates both single uridines and short ssRNA fragments as degradation products that bind TLR8 and initiate signal transduction [20,21,22]. It was also reported that 2′-O-ribose methylation of phosphodiester ssRNA impairs RNase digestion, and thus, influences TLR8 activation [20]. This observation necessitates further research on the influence of 2′-O-methylations on TLR8 activity, as the importance of therapeutic TLR8 ligands has greatly increased in recent years [23,24,25,26]. For this purpose, the use of 2′-O-ribose methylation may be useful, as it can both stabilize RNA and lead to adequate control of the immune response.

Several groups have reported that RNA methylations can prevent activation of TLR7 [27,28,29,30]. Remarkably, a single 2′-O-ribose methylation is sufficient to prevent TLR7 signaling and convert a TLR7/TLR8 ligand into a sole TLR8 ligand; however, the influence of the positioning of 2′-O-methylation has not yet been explored in much detail [31,32,33]. Therefore, it is important to determine whether the position of 2′-O-methylation matters, as it may be of both technical and immunological relevance for the most efficient generation of TLR8 ligands.

The aim of this study was to investigate whether a known TLR8 ligand could be further optimized by alternating the position of the 2′-O-ribose methylation. Taking the importance of RNase activity into account, 2′-O-ribose methylation may alter the preferred RNA cleavage site to different positions, thus generating RNA degradation products with strong differential effects on TLR7 and TLR8 activity. As TLR7 activation should be avoided, we focused only on guanosine methylations, because binding of both single guanosine nucleosides and guanosine-rich sequences is essential for TLR7 activation [14,16,18,19,34]. To this end, we designed different oligoribonucleotides (ORNs) that were methylated at different guanosine positions and derived from different naturally occurring RNA sequences. Following RNase digestion, we monitored the effect of 2′-O-methylations on RNA degradation patterns. TLR8 activation by methylated and unmethylated ORNs was investigated by immunostimulations with primary human immune cells, murine pDCs, and TLR8-transfected HEK293 reporter cells. We observed a flexibility in TLR8 activation with respect to the position of 2′-O-methylation on its ligands. The results presented here contribute to the generation of optimal TLR8 ligands under different requirements and for diverse clinical applications.

## 2. Results

### 2.1. 2′-O-Ribose Methylation Prevents TLR7 Activation Independent of the Position

We aimed to investigate whether the conversion of a TLR7 and TLR8 ligand into a sole TLR8 ligand was a universal effect of 2′-O-ribose methylation or whether this effect was mediated only by a small group of naturally occurring methylation patterns, located at specific positions. Therefore, we first employed the previously characterized TLR7 and TLR8 ligand, RNA63, derived from 18S rRNA (position 1488–1499), as its immunostimulatory properties have already been confirmed in previous publications (Figure 1a) [30,32]. Under natural conditions, this ligand is 2′-O-ribose methylated at the first guanosine (RNA63M1) [35]. To investigate the influence of the position of the methyl group on preferential RNase cleavage sites and immune activation, we designed four different RNA63 derivatives, each with a 2′-O-ribose methylation at one of the four guanosines (RNA63M1 to RNA63M4). RNaseT2 activity has been shown to be necessary for TLR8 activation and affected by 2′O-methylation [20,21]. Thus, we compared methylated and unmethylated RNA63 with and without RNase T2 treatment and observed that RNA methylation indeed resulted in altered RNA fragmentation patterns (Figure 1b and Figure A1). In particular, RNase T2-digested RNA63M1 and RNA63M3 showed a slightly differential fragment pattern compared with RNA63M2 and RNA63M4. Specifically, RNase digests of RNA63M1 and RNA63M3 demonstrated at least five fragments with a prominent double band (marked by an arrow 
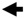
). In contrast, the analysis of RNA63M2 and RNA63M4 digests showed an altered pattern in this region resembling a single band or two co-migrating bands. Furthermore, digested RNA63M1 and RNA63M3 showed two additional small bands (compared with a single smaller band in digested RNA63M2 and RNA63M4), which is marked with a line (

). Overall, these patterns suggest a difference in ORN fragments created by RNase T2 digestion.

It has been reported that human TLR7 induces both IFN-α and interleukin-6 (IL-6) release, whereas human TLR8 leads to IL-6 release but not IFN-α [5,15,36,37]. We then tested whether 2′-O-methylations at different positions on the RNA would lead to changes in the observed cytokine response. To this end, we stimulated human peripheral blood mononuclear cells (PBMCs) with the different ORNs, and both IFN-α and IL-6 release was measured by ELISA (Figure 1c,d). The previously described unmethylated oligodeoxynucleotide (CpG ODN 2216), which is a well-known TLR9 ligand that contains CG dinucleotide, was used as a TLR7-independent positive control for pDC activation [6,7,38]. RNA40, a well characterized TLR7 and TLR8 ligand, was used as a specific TLR7/8 control agonist [14]. In human PBMC, CpG2216 induced IFN-α release, but not IL-6. This is consistent with previously descriptions [38] of the functional activity of the pDCs present within the isolated PBMC [6,7]. In contrast, RNA40 induced both IFN-α and IL-6 release, in line with previous studies on TLR7 and TLR8 in human pDCs and monocytes [14]. In contrast, we observed that while RNA63 induced IFN-α secretion, each 2′-O-ribose-methylation that was applied strongly decreased IFN-α release (Figure 1c). In contrast, RNA63M1-M4 derivatives induced a robust IL-6 signal (Figure 1d). Comparing the differentially methylated RNA63 derivatives, we observed that RNA63M2 and RNA63M4 still induced slight IFN-α release (Figure 1c) and caused a slightly weaker IL-6 induction than both RNA63M1 and RNA63M3 (Figure 1d). Nonetheless, our data suggest that human TLR7 activation can be inhibited by guanosine methylation in RNA63, irrespective of the position, whereas TLR8 activation remains unaffected

In mice, TLR7 can also induce both IFN-α and IL-6 release, whereas TLR8 activation does not induce these cytokines [9,10,12,13]. To examine murine TLR7 activity, we generated FMS-like tyrosine kinase 3 ligand-differentiated dendritic cells (FLT3L DCs) from the bone marrow of wildtype (wt) and Tlr7-deficient (*Tlr7*^−/−^) mice and stimulated these cells with both RNA63 and methylated RNA63M1-M4 derivatives (Figure 2). As in human PBMC, induction of both IFN-α and IL-6 secretion by CpG ODN 2216 confirmed the activity and responsiveness of FLT3L DCs, in both wildtype and *Tlr7^−/−^,* to nucleic acid stimuli. In contrast, IFN-α and IL-6 secretion was completely abrogated in *Tlr7^−/−^* after RNA40, demonstrating the expected functional activity of this ligand in mice.

Whereas RNA63 induced a significant IFN-α and IL-6 response in FLT3L DCs of wt mice, the Tlr7 knockout completely abolished RNA63-dependent cytokine secretion (Figure 2). Of note, each 2′-O-ribose-methylation of RNA63 abrogated both IFN-α and IL-6-release. Consequently, murine TLR7 activation was inhibited by guanosine methylations of RNA63, independent of the position of the modification.

### 2.2. Impact of 2′-O-Ribose Methylation on TLR7 Activation by Naturally Methylated and Unmethylated Sequences

We then investigated whether the generation of a strong TLR8 ligand could only be achieved by 2′-O-methylation of specific RNA sequences such as the optimized RNA63, or whether this effect could be universally achieved using GU-containing ORNs. We used two additional RNA sequences in their unmethylated and methylated derivatives: RNA28, which is derived from 28S rRNA (position 2409–2420 on 28S rRNA, XR_007090848.1) and carries a 2′-O-methylation at the first guanosine under natural conditions, and RNA66, which is derived from 18S rRNA (position 773–784 on 18S rRNA, NR 003286.2) and is not methylated under natural conditions (Figure 3a) [35]. The TLR7/TLR8 stimulatory activities of these RNA sequences have not been previously described, and these particular sequences were selected due to their high uridine content, natural occurrence in ribosomal RNA, and presence of a guanosine at position 3. PBMCs were stimulated with the ORNs, and RNA-induced cytokine release was determined by ELISA (Figure 3b,c).

As shown in Figure 1, CpG2216 induced IFN-α, providing a TLR7-independent control of pDC activity. Moreover, RNA40, as well as both RNA28 and RNA66, induced IFN-α and IL-6 release, as was observed for RNA63 (Figure 3b,c, compare Figure 1). In contrast, RNA28M induced the release of IL-6 only, as observed for the methylated derivatives of RNA63, whereas RNA66M did not induce IFN-α or IL-6 release, indicating that this particular 2′-O-methylation blocked both TLR7 and TLR8 activation.

To examine RNA28- and RN66-induced immune activation in murine cells, we studied TLR7 using wt and *Tlr7*^−/−^ murine dendritic cells.

In murine FLT3L DCs, as shown in Figure 2, CpG2216 and RNA40 acted as TLR7-independent and TLR7-dependent controls, respectively. However, both RNA28 and RNA66 were weak agonists of TLR7 in comparison with RNA40 and RNA63 (Figure 4a, compare Figure 2). Nonetheless, IFN-α release was induced by RNA28 and RNA66 in a TLR7-dependent manner. Moreover, in line with what was observed for RNA63M1-4, neither RNA28M nor RNA66M induced IFN-α (Figure 4a). RNA28 also only induced low levels of IL-6, which were inhibited by 2′-O-methylation (RNA28M), as seen for RNA63M1-4. However, RNA66 stimulation did not induce IL-6 release from murine FLT3L DCs (Figure 4b).

### 2.3. Activation of TLR8 by 2′-O-Ribose Methylated ORNs

Our previous data demonstrating IL-6 but not IFN-α induction after stimulation with 2′-O-ribose methylated ORNs only provide an indirect indication that TLR8 is activated by 2′-O-ribose methylated ORNs. To provide direct evidence of TLR8 activation, we used a HEK reporter cell line stably overexpressing TLR8 (HEK-TLR8) and stimulated it with the different methylated and unmethylated ORNs.

Immune stimulation of TLR8 reporter cells resulted in activation of the transcription factor nuclear factor ′kappa-light-chain-enhancer′ of activated B-cells (NF-κB), which resulted in the release of secreted embryonic alkaline phosphatase (SEAP) and was normalized to the SEAP activity induced by the synthetic TLR8-specific agonist TL8-506 [39] (Figure 5). Although mock treatment of cells (mock) without ORN did not lead to NF-κB-activation, transfection of all methylated RNA63 derivatives upregulated SEAP activity (Figure 5a). However, the strength of immune activation differed between the individual RNA63 derivatives, and 2′-O-methylation at position 3 (RNA63M1) resulted in the weakest signal transduction in TLR8 reporter cells. In contrast, TLR8 activation by the other ORNs was different; although RNA66 and RNA28 led to distinct TLR8 activation, RNA66M and RNA28M only led to weak NF-κB activation (Figure 5b,c).

In summary, a single 2′-O-methylation, tested at different positions within the RNA, converted RNA63 from a TLR7/TLR8 ligand into an exclusive TLR8 ligand. However, 2′-O-methylations at different guanosine positions resulted in different strengths of TLR7 inhibition and TLR8 activation, although it was observed for all positions tested. In addition, the switch from TLR7 and TLR8 activation to sole TLR8 activation worked particularly well for the optimized RNA63 derivatives and seemed to be a coordinated process, as some methylated sequences showed only weak TLR8-activating properties.

## 3. Discussion

The application of RNA as a therapeutic agent has become increasingly important in recent years [23,24,25,26]. Just as important as the production of stable ORNs that lead to sufficient immune activation is the targeting of the innate immune response to avoid an excessive or misdirected danger signal. For this reason, it is essential to design optimal PRR-ligands and gain knowledge about the optimal localization of ORN methylation, especially for cases with technical limitations. Consequently, we aimed to design an optimized TLR8 ligand by investigating the influence of 2′-O-methylations at different positions on TLR8 activity. We first investigated different derivatives of the previously described ORN RNA63 [32]. This sequence was converted from a TLR7/TLR8 ligand to a TLR8 ligand by a single ribose methylation at position 3. We designed several RNA63 derivatives that were methylated at different guanosine positions and applied them in PBMC stimulations (Figure 1). All RNA63 derivatives with a single ribose methylation strongly decreased or abrogated IFN-α release from human PBMCs but still induced robust IL-6 release. Of note, RNA63M2 and RNA63M4, which showed residual IFN-α induction and weaker IL-6 release from human PBMCs, also exhibited a different RNA fragmentation pattern than RNA63M1 and RNA63M3 derivatives, which induced a strong IL-6 signal only. Therefore, we speculate that a single 2′-O-methylation may indeed direct RNase activity, thereby aiding the development of optimal TLR8 ligands essential for TLR8 activation [20,21,22].

To demonstrate the prevention of TLR7-dependent immune activation by 2′-O-ribose methylation, we stimulated DCs differentiated from the bone marrow of both wt and *Tlr7^−/−^* mice (Figure 2). These experiments proved that unmodified RNA63 is a TLR7 ligand, and TLR7 activity was inhibited by 2′-O-methylation at all positions tested. Thus, we demonstrated the flexibility of TLR7 inhibition with respect to the position of 2′-O-methylation.

As our data did not allow us to draw conclusions about whether the conversion from TLR7 to TLR8 by 2′-O-methylation was transferable to other sequences, we designed two additional ORNs, RNA28 and RNA66, as well as their methylated derivatives. The RNA28 sequence, similar to RNA63, is methylated under natural conditions [35]. Unmethylated RNA28 appears to be only a weak TLR7 activator, as stimulation of both human PBMCs and murine FLT3L-differentiated DCs with RNA28 resulted in only weak IFN-α release. Nevertheless, RNA28M proved to be a good TLR8 inducer in primary human immune cells, as it led to a comparably strong IL-6 induction as its unmethylated derivate and RNA63. In contrast, RNA66 but not RNA66M led to marked cytokine release. This provided initial evidence that RNA66M is not a strong TLR8 ligand.

We used murine FLT3-differentiated DCs to confirm the TLR7 dependence of RNA28 and RNA66. Indeed, both sequences were shown to be TLR7 ligands (Figure 4). The observation of weak RNA28-mediated cytokine release in the murine system supports the hypothesis that RNA28 is a low-potency TLR7 activator (compare Figure 3). RNA66, on the other hand, was a strong IFN-α inducer in the murine system, as it is in humans, but did not induce IL-6. At this point, the question arises why the RNA66 sequence, which was already a weak IL-6 inducer in human PBMCs and is not methylated under natural conditions [35], was not recognized as a pattern that requires a proinflammatory danger signal. For this reason, it seemed necessary to directly verify which RNA sequences were TLR8 ligands.

Therefore, we tested the TLR8 specificity of the different RNA sequences in TLR8-expressing HEK reporter cells (Figure 5). Indeed, all unmethylated and methylated ORNs induced significant TLR8-dependent NF-κB activation, which was weak in the cases of RNA63M1, RNA66M, and RNA28M. Nevertheless, we assume that RNA63M1 and RNA28M are TLR8 ligands because they induced a robust IL-6 response in PBMCs (compare Figure 1). In contrast, RNA66M-induced IL-6 release was significantly reduced in PBMCs compared with RNA66, so we hesitate to conclude that RNA66M is also a TLR8 ligand. With respect to our objective of identifying an optimal and specific TLR8 ligand, we would choose the newly designed RNA63M3. the previously described RNA63M1 would be a second choice, as it showed no TLR7 activity and robust TLR8 activity in all systems tested. The fact that randomly selected ORNs such as RNA28M and RNA66M only had weak TLR8 activity or lacked it completely highlights the importance of RNA63M3.

The differences in TLR7 and TLR8 activity of RNAs methylated at different positions are probably because 2′-O-methylation can protect against RNase digestion of the phosphodiester bond between the methylated and 3′ nucleotide [20,40]. In our case, 2′-O-methylation occurred at only a single variable position, leaving wide stretches of ssRNA accessible to RNase digestion, which was reported to be essential for TLR8 activation [20,21,22]. However, 2′-O-methylations at different positions within the RNA most likely also directed preferential RNA cleavage to different positions, resulting in an RNA fragmentation pattern that was also variable between individual RNA63 derivatives. Consequently, both short single-stranded RNA stretches and single nucleosides reported to be essential for TLR7 and TLR8 activation can be generated from all methylated derivatives, but methylation at the optimal site presumably results in the generation of RNA degradation products that activate TLR8 and not TLR7 [18,19,20,21,22,34]. Here, the base composition of the resulting RNA fragments and their methylation status both play a role, as 2′-O-methylations have been reported to prevent TLR7 and TLR8 activation by ssRNA [20,30,41]. However, a balance between stability and accessibility to RNases seems to be required, as it has also been reported that some ORNs lead to better TLR7 and TLR8 activation when made more resistant to nuclease-mediated digestion by modification [42].

Consistent with our data, specific TLR8 activation by ORNs with single methylations has been reported previously [27,31,32]. However, these publications only investigated naturally occurring 2′-O-methylations, and not the influence of position, which we have investigated here.

To verify the specificity of our potential optimal TLR8 ligands in our systems, we also examined other naturally methylated and unmethylated RNA segments for their TLR7 and TLR8 activity. An influence of the sequence context of the 2′-O-methylation on the silencing of TLR7 and TLR8 has already been discussed in previous publications [28,29]. In this context, we observed that RNA66, compared with RNA63 and RNA28, was no longer a strong TLR8 ligand after ribose methylation. However, unlike RNA63 and RNA28, RNA66 does not contain AU segments, which were reported to be important for a TLR8 ligand [16]. However, it should be noted that all ORNs tested (including RNA66) contained GU segments, which have been described to induce TLR8 activation [14]. Of note, we observed strong TLR8 activity after ribose methylation in ORNs RNA63M1–M4 and RNA28M. Therefore, we conclude that only RNA sequences which are strong TLR8 ligands per se still induce NF-κB activation after a single 2′-O-methylation.

In summary, we have shown that a single 2′-O-methylation at any position can convert a TLR7/8 ligand to an exclusive TLR8 ligand, but with varying efficiency. Here, the RNA63M3 derivative turned out to be a particularly suitable TLR8 ligand, as 2′-O-methylation at other positions or other methylated sequences resulted in less potent TLR8 ligands or less potent inhibition of TLR7 signaling. These findings could facilitate the generation of therapeutic TLR8 ligands.

## 4. Materials and Methods

### 4.1. Kits and Reagents

TL8-506 was purchased from Invivogen (Toulouse, France) and RNA40 (GCCCGUCUGUUGUGUGACUC) was purchased from IBA (Göttingen, Germany). Phosphodiester-linked ORNs with the following sequences were synthesized by Metabion (Planegg, Germany): RNA63 (caggucugugau), RNA63M1 (caxgucugugau; “x” depicts 2′-O-methyl-guanosine), RNA63M2 (cagxucugugau), RNA63M3 (caggucuxugau), RNA63M4 (caggucuguxau), RNA28 (caguugaacaug), RNA28M (caxuugaacaug), RNA66 (cugagugucccg), and RNA66M (cuxagugucccg). CpG ODN 2216 was obtained from Biomol (Berlin, Germany). Poly-dT-phosphorothioate (PTO) was provided by Metabion (Martinsried, Germany). Recombinant human FMS-like tyrosine kinase 3 ligand (FLT3L) was obtained from H. Hochrein (Bavarian Nordic GmbH, Martinsried, Germany).

### 4.2. Cells

Human peripheral blood mononuclear cells (PBMCs) were purified from buffy coats by Ficoll gradient centrifugation (PAA Laboratories GmbH, Cölbe, Germany) and cultivated in RPMI1640 (PAN Biotech GmbH, Aidenbach, Germany), supplemented with 2 mM glutamine (Merck, Darmstadt, Germany), 100 units/mL penicillin and 100 µg/mL streptomycin (Merck), non-essential amino acids (Merck), 1 mM sodium pyruvate (Merck), and 2% serum of AB positive male donors (Merck). Differentiation of dendritic cells (DCs) from the bone marrow of both wt and *Tlr7^−/−^* mice was induced by stimulation with FLT-3 ligand, as described in [43], and cells were maintained in Opti-MEM (Thermo-Fisher, Waltham, Massachusetts) with 10% fetal calf serum (FCS) (Biochrom AG, Berlin, Germany), 2 mM glutamine, 100 units/mL penicillin, 100 µg/mL streptomycin, and 0.05 mM β-Mercaptoethanol (Sigma-Aldrich, St. Louis, MO, USA).

### 4.3. Cell Stimulation

For cell stimulation, PBMCs were seeded at 3 × 10^5^ cells/well and murine DCs were seeded at 2 × 10^5^ cells/well in 100 µL growth medium in a 96-well flat-bottom plate. Cells were stimulated with CpG ODN 2216 at 1 µM without a transfection reagent as stimulation control. For stimulation with ORNs, RNA40 was applied at 5 µg/mL, and RNA63, RNA66 and RNA28 ORNs were applied at 10 µg/mL. The stimulation mixture was prepared as follows: RNA stimuli were combined with 50 µL Opti-MEM and 1.5 µL DOTAP (Roche, Mannheim, Germany) per well. Samples were incubated for 5 min at room temperature, then 50 µL of the medium was added without serum and used to stimulate cells at 100 µL. Supernatants were harvested 20 h post stimulation (h p. s.) and cytokine secretion was determined by ELISA analysis.

### 4.4. Enzyme-Linked Immunosorbent Assay (ELISA)

Murine IL-6, murine IFN-α, and human IL-6 concentrations were measured by ELISA according to the manufacturer′s instructions (R&D Systems, Minneapolis, MN, USA for murine IL-6; PBL Interferon Source, Piscataway, NJ, USA for murine IFN-α; BD Biosciences, Heidelberg, Germany for human IL-6). Human IFN-α concentrations were measured using capture antibody mouse monoclonal anti-human IFN-α (PBL Interferon Source) and detection antibody anti-human IFN-α HRP-conjugate (eBioscience, San Diego, CA, USA). Recombinant human IFN-α (PeproTech, Cranbury, NY, USA) was used as a standard.

### 4.5. Genetic Complementation Assay

HEK-Blue™ hTLR8 (Invivogen) cells expressing the human TLR8 gene and a NF-κB/AP-1 inducible SEAP (secreted embryonic alkaline phosphatase) reporter gene were used as a TLR8-specific reporter system. Upon TLR8-dependent NFκB activation, SEAP activity was monitored using QUANTI-Blue™ (Invivogen). Cells were seeded in a 96-well flat-bottom plate at 6 × 10^4^ cells/well. Cells were stimulated 12 h after seeding with final concentrations of 0.5 µg/mL TL8-506 or 5 µg/mL RNA40, and 20 µg/mL RNA63, RNA66, and RNA28 ORN complexed to 5 µL DOTAP/well. To enhance TLR8-activation, Poly-dT-PTO was added at 1.5 µM final concentration. The readout was performed with QUANTI-Blue™ medium, as described in the manufacturer′s protocol. Optical density was measured with a Berthold luminometer (Pforzheim, Germany) at a wavelength of 650 nm. Results were normalized to percent NF-κb induction by TL8-506.

### 4.6. Mice

For experiments with Tlr7-deficient mice, we used the Tlr7-deficient mouse line established by Hemmi et al. [12]. Mice were bred under specific pathogen-free conditions at the animal facility of Philipps University Marburg.

### 4.7. RNase T2 Digestion Assay

An amount of 500 ng of each ORN was incubated with 0.0025 U RNase T2 (Aspergillus oryzae, MoBiTec GmbH, Göttingen, Germany) for 1.5 h at 37 °C. RNA fragments were separated in a 25% polyacrylamide gel with the following running conditions: 3–4 h at 250 V, 8 °C. Staining of RNA bands was performed in a 0.05% toluidine blue aqueous solution with 20% methanol and 2% glycerol for 1 h. Destaining and visualization of band patterns was performed in a destain solution containing 20% methanol and 2% glycerol for at least 2 h. Documentation was performed on a Chemi Doc Imaging System (Bio-Rad, Hercules, CA, USA).

### 4.8. Statistical Analysis

Experiments tested for statistical significance were performed six times. Data were tested for normality using D′Agostino-Pearson and Kolmogorov-Smirnov tests and are presented as arithmetic means + S.D. Statistical analyses of normally distributed data were based on paired two-tailed *t*-tests. Non-normally distributed data were analyzed using Wilcoxon tests.

## Figures and Tables

**Figure 1 ijms-23-11139-f001:**
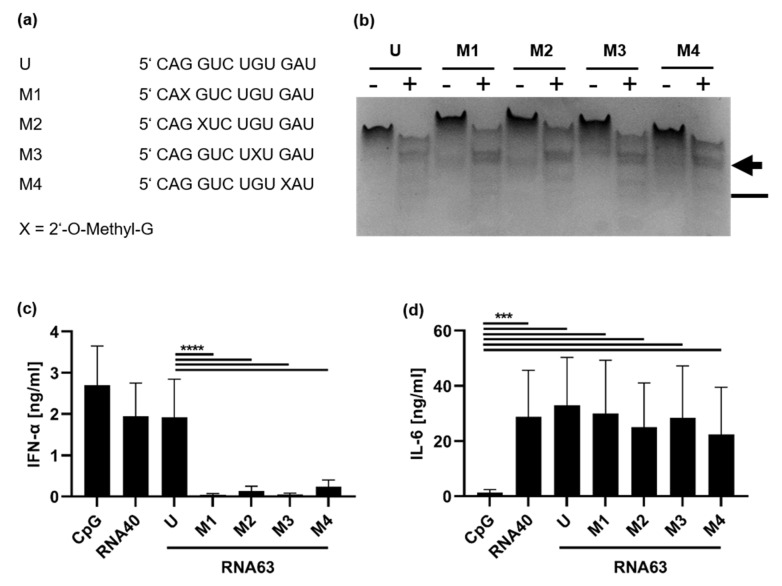
Methylation by 2′-O-ribose prevents IFN-α but not IL-6 secretion, independent of the position. (**a**) Sequences of unmethylated RNA63 (U) or methylated derivatives (M1, M2, M3, and M4): 2′-O-ribose-methylated guanosines are indicated with an X. (**b**) RNA63 derivatives were either treated with 0.005 units of RNase T2/µg RNA (+) or mock treated (−), and RNA fragmentation was visualized in polyacrylamide gel electrophoresis. Position of the RNA fragments of interest are indicated with an arrow (
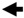
) and a line (

). (**c**,**d**) PBMCs were stimulated with RNA63 derivatives at a final concentration of 10 µg/mL: 1 µM CpG ODN 2216 or 5 µg/mL RNA40. Supernatants were harvested 20 h post-stimulation (h p. s.) and both (**c**) IFN-α and (**d**) IL-6 concentrations were measured in ELISA. Graphs depict six independent experiments with PBMCs obtained from six individual donors, each in biological duplicates (twelve measurements per data point, mean + S.D). Data were analyzed using paired *t*-tests. **** *p* < 0.0001, *** *p* < 0.001.

**Figure 2 ijms-23-11139-f002:**
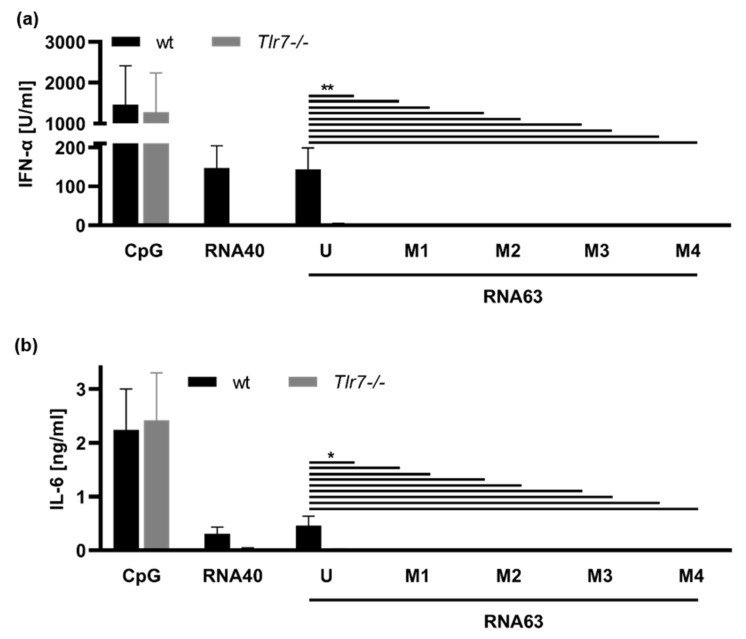
Methylation by 2′-O-ribose of a different guanosine prevents TLR7 activation. FLT3L DCs from wt or *Tlr7*^−/−^ mice were stimulated with 1 µM CpG ODN 2216 and 5 µg/mL RNA40 or RNA63 derivatives (U, M1, M2, M3, or M4) at a final concentration of 10 µg/mL. Supernatants were harvested 20 h p. s. and concentrations of (**a**) IFN-α and (**b**) IL-6 were determined by ELISA. Graphs depict six independent experiments each in biological duplicates (twelve measurements per data point, mean + S.D). Data were analyzed using Wilcoxon tests. ** *p* < 0.01, * *p* < 0.05.

**Figure 3 ijms-23-11139-f003:**
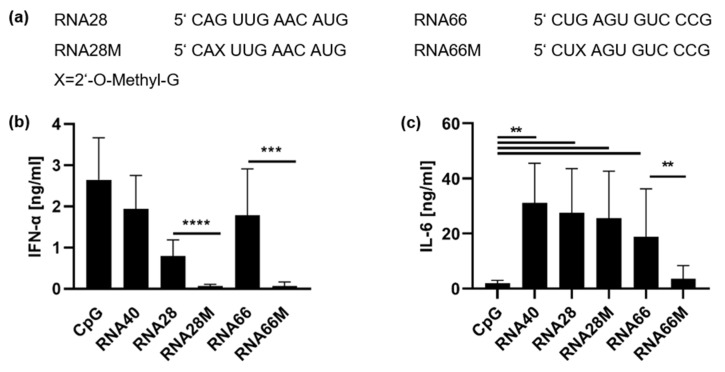
Impact of 2′-O-ribose methylation on cytokine induction by naturally methylated and unmethylated sequences. (**a**) Sequences of applied RNA derivatives. 2′-O-ribose methylated guanosine are indicated with an X. (**b**,**c**) PBMCs were stimulated with RNA66 or RNA28 derivatives at a final concentration of 10 µg/mL, with 1 µM CpG ODN 2216 or 5 µg/mL RNA40. Supernatants were harvested 20 h p. s. and both (**b**) IFN-α and (**c**) IL-6 concentrations were measured by ELISA. Graphs depict six independent experiments with PBMCs obtained from six individual healthy donors each in biological duplicates (twelve measurements per data point, mean + S.D). Data were analyzed using paired t-tests; IL-6-data of RNA66 and RNA66M were analyzed using Wilcoxon tests. **** *p* < 0.0001, *** *p* < 0.001, ** *p* < 0.01.

**Figure 4 ijms-23-11139-f004:**
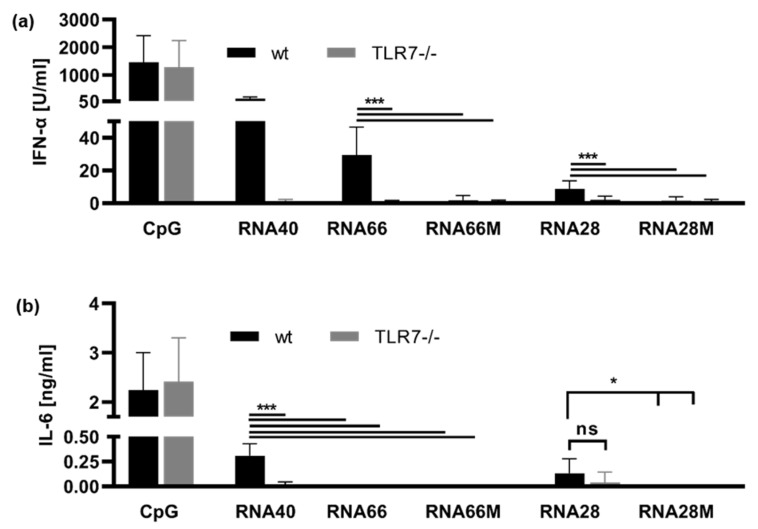
Methylation by 2′-O-ribose of different sequences prevents TLR7 activation. FLT3L DCs from wt or *Tlr7*^−/−^ mice were stimulated with 1 µM CpG ODN 2216, 5 µg/mL RNA40, and RNA66 derivatives or RNA28 derivatives at a final concentration of 10 µg/mL. Supernatants were harvested 20 h p. s. and concentrations of (**a**) IFN-α and (**b**) IL-6 were determined in ELISA. Graphs depict six independent experiments each in biological duplicates (twelve measurements per data point, mean + S.D). Data were analyzed using Wilcoxon tests. *** *p* < 0.001, * *p* < 0.05, ns = not significant.

**Figure 5 ijms-23-11139-f005:**
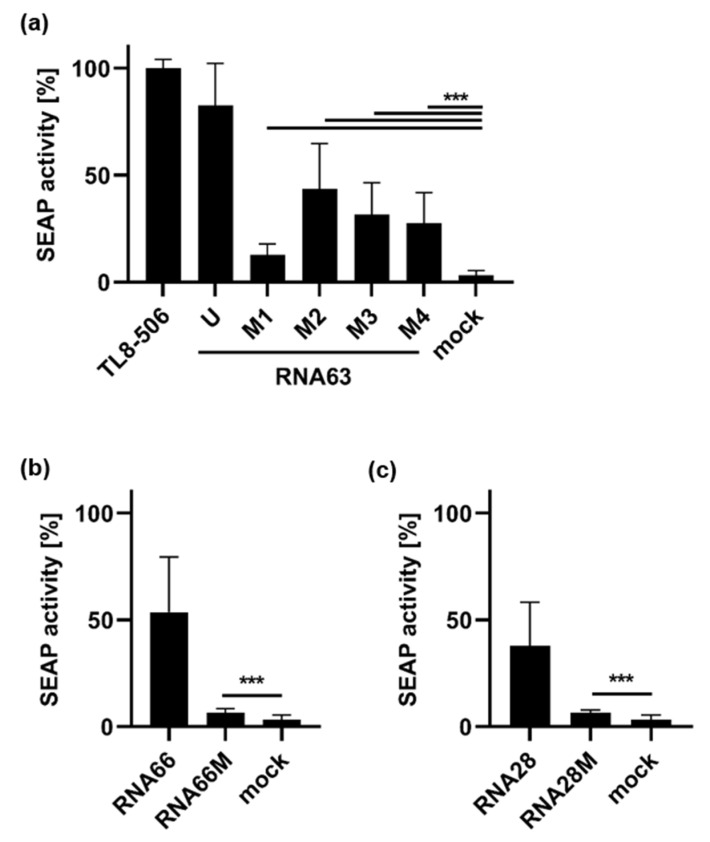
Oligoribonucleotides with 2′-O-ribose methylation activate TLR8. HEK-TLR8 cells were stimulated with TLR8-specific TL8-506 at 0.5 µg/mL and different derivatives of (**a**) RNA63 (U = unmethylated), (**b**) RNA66, or (**c**) RNA28 at a final concentration of 20 µg/mL or mock-treated. Cell supernatant was mixed with QUANTI-Blue™ medium at 20 h p. s. for 15 min at 37 °C. Secreted embryonic alkaline phosphatase (SEAP) activity, reflecting NF-κB activation, was determined by colorimetric measurement at 650 nm and results were normalized to TL8-506-induced NFκB activation. Graphs depict six independent experiments each in biological duplicates (twelve measurements per data point, mean + S.D). Data were analyzed using Wilcoxon tests. *** *p* < 0.001.

## Data Availability

Not applicable.

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
