# Peer review of "Identification of an Optimal TLR8 Ligand by Alternating the Position of 2′-O-Ribose Methylation"

_ijms, 2022, doi:10.3390/ijms231911139_

Round 1
Reviewer 1 Report
The manuscript by Nicolai et al. entitled “Identification of an optimal TLR8 ligand by alternating the position of 2'-O-ribose methylation” investigates the impact of 2'-O-ribose methylation on a number of RNA oligos’ ability to activate TLR7 and TLR8. Multiple systems, including human PBMCs, murine DCs, and HEK293-based TLR8 reporter line, are used to directly or indirectly test these oligos with or without methylation. The data presented in the manuscript are largely clear and convincing. The following concerns and suggestions can be addressed to improve the manuscript.
1. Throughout the manuscript, there is a lack of explanation of experimental controls used (e.g. CpG, RNA40, TL8-506, and the “¢” in Fig 5) as well as the anticipated results from each control. These are vital components of the experiments and should be explained in the Results section.
2. Are all the PBMCs used in this study isolated from the same donor? If so, using PBMCs from at least another donor would strengthen the conclusions and reduce potential biases from a single host’s genetic background.
3. Statistical analysis results are not always shown or properly labeled. For example, in Fig 1c, the line with **** should only indicate that U is significantly different from M4, while the main text indicates that U is different from M1-4. There are multiple examples like this in the manuscript that should be corrected.
4. What is the reporter gene expressed by HEK-TLR8 and how is NFkB activation determined by colorimetric measurement? A very brief explanation of the reporter mechanism would help.
5. The HEK-TLR8 provides more direct measurements of human TLR8 activation, while there is not a system like this for human TLR7 in this study to compare how much methylation reduces its activation. Can the authors test these unmethylated vs. methylated RNA oligos in a human TLR7 reporter cell line?
Reviewer 2 Report
Nicolai et al in their manuscript entitled “Identification of an optimal TLR8 ligand by alternating the position of 2’-O-ribose methylation” aimed to investigate how 2’-O-methylation converts TLR7/TLR8 ligands to TLR8-specific ligands. My guess is the ultimate goal was to develop a tool for proper design of TLR-specific ligands. The authors were able to confirm that 2’-O-methylation at a single position in an RNA oligo already known as a TLR7/TLR8 ligand can convert this ligand to TLR8-specific (RNA63). Another RNA oligo the authors designed and tested was found a very weak ligand for TLR8 when it was methylated (RNA66). Thus, the authors could not provide any useful insights into optimal TRL8 ligand design.
Overall the paper is not well-written. It is full of abbreviations and acronyms that are not spelled out at all or it is done at the end of the manuscript: DC, Flt3, NF-kB, PBMC etc.
Reagents used in the study are poorly described. What is TL8-506? What does CpG stand for? If it is CpG ODN2216 designed and verified as TLR9 ligand in a different paper that must be clearly stated and credits must be given to the authors of that paper. Why this ligand is used in the present study is not explained anywhere in the text. Why particular fragments of human rRNAs were chosen to test in this and previous studies (RNA63, RNA28, RNA66) is not obvious. Were they picked at random? What is RNA40? The sequence provided in the methods section is not helpful to understand the use of this RNA oligo.
Authors claim that they ‘constructed’ RNA oligonucleotides used in the study and call synthetic RNA oligos ‘constructs’. In fact, they just ordered RNA oligos either unmodified or 2’-O-methylated from companies that provide oligo synthesis service.
I wonder if the authors made tlr7-/- mice themselves again using the procedure described in the cited paper. It sounds like that in section 4.6. of Methods (line 354).
Generally speaking, Quanti-blue assay measures alkaline phosphatase activity. In the context of this paper, the assay quantifies NF-kB-induced secreted alkaline phosphatase not NF-kB activation per se.
I do not understand why 3 gels provided in supplementary material show different patterns of the same experimental set up: unmodified and modified RNA oligos mock treated or treated with RNase T2. By the way, why this enzyme was used needs an explanation in the text. It seems that one replicate was a total failure, and all 3 are not good quality gels. Then, for figure 1b, the authors picked one that matched better with the pattern of IFN-alpha activation (Figure 1c). This is not acceptable.
These are just few examples, not an explicit list.
Round 2
Reviewer 2 Report
The revised version of this manuscript shows significant improvement. Though the quality of the gels showing RNase T2 digestion patterns is still questionable (the major concern I had originally) additional images and descriptions make these data more convincing. If the authors plan to provide these images as supplementary data for the paper, they should update the originally uploaded file.
I have few comments and suggestions that might help further improve the paper.
I think the authors should not call a switch from TLR7/TLR8 activation to TLR8 only “TLR7 to TLR8 conversion” (see line 250) or “conversion of a TLR7 into a TLR8 ligand” (see line 81). They cannot uncouple TLR7 and TLR8 activation by unmodified RNA oligos tested in this study.
I would not call RNase T2 treatment ‘RNase T2 degradation assay’ (line 417), it is rather ‘RNase T2 digestion assay’.
Figure legends contain redundant description of methods, yet do not explain what exactly those plots show.
Line 115, 154, 184, 201, 227. I guess bars in Figures 1, 2, 3, 4 and 5 show ‘average’ of 12 measurements +/- SD.
Some grammar and style corrections are still required.
e.g., in line 40, ‘also’ is redundant; in line 125, I would say “a TLR7-independent positive control”, etc.
I believe MDPI journals provide this type of editing prior to publication.
Author Response
Thank you for your kind message and positive feedback. We have been glad to incorporate your helpful suggestions for changes to further improve the manuscript.
The revised version of this manuscript shows significant improvement. Though the quality of the gels showing RNase T2 digestion patterns is still questionable (the major concern I had originally) additional images and descriptions make these data more convincing. If the authors plan to provide these images as supplementary data for the paper, they should update the originally uploaded file.
Thank you for this helpful suggestion. The additional gel images were integrated into the manuscript as supplementary data A1.
I have few comments and suggestions that might help further improve the paper.
I think the authors should not call a switch from TLR7/TLR8 activation to TLR8 only “TLR7 to TLR8 conversion” (see line 250) or “conversion of a TLR7 into a TLR8 ligand” (see line 81). They cannot uncouple TLR7 and TLR8 activation by unmodified RNA oligos tested in this study.
Thank you for this good comment. The relevant passages have been modified.
I would not call RNase T2 treatment ‘RNase T2 degradation assay’ (line 417), it is rather ‘RNase T2 digestion assay’.
You are absolutely right that this term is much more accurate. The sentence in question was changed according to your suggestion.
Figure legends contain redundant description of methods, yet do not explain what exactly those plots show.
Line 115, 154, 184, 201, 227. I guess bars in Figures 1, 2, 3, 4 and 5 show ‘average’ of 12 measurements +/- SD.
Thank you for this helpful comment. Indeed, the graphs show the mean +/- standard deviation. This has now been included in the manuscript.
Some grammar and style corrections are still required.
e.g., in line 40, ‘also’ is redundant; in line 125, I would say “a TLR7-independent positive control”, etc.
I believe MDPI journals provide this type of editing prior to publication.
Thank you for these conscientious comments, we have modified the relevant passages. To ensure a correct language, the entire manuscript was again subjected to a further complete language check by a native English speaker.